# Chemical Composition, Antibacterial, Anti-Inflammatory, and Enzyme Inhibitory Activities of Essential Oil from *Rhynchanthus beesianus* Rhizome

**DOI:** 10.3390/molecules26010167

**Published:** 2020-12-31

**Authors:** Xiaoge Zhao, Qi Chen, Tingya Lu, Feng Wei, Yao Yang, Dan Xie, Huijuan Wang, Minyi Tian

**Affiliations:** 1National & Local Joint Engineering Research Center for the Exploition of Homology Resources of Southwest Medicine and Food, Guizhou University, Guiyang 550025, China; zhaoxiaoge123@163.com (X.Z.); ChenQi272727@163.com (Q.C.); lutingya1602491445@163.com (T.L.); weifeng13765070634@163.com (F.W.); yangyao2821@163.com (Y.Y.); xiedan1630210@163.com (D.X.); xuchwhj@163.com (H.W.); 2College of Life Sciences, Guizhou University, Guiyang 550025, China

**Keywords:** *Rhynchanthus beesianus* W. W. Smith, essential oil, GC-MS, antibacterial activity, anti-inflammatory activity, enzyme inhibitory activity

## Abstract

*Rhynchanthus beesianus* W. W. Smith, an edible, medicinal, and ornamental plant, is mainly cultivated in China and Myanmar. The essential oil (EO) from *R. beesianus* rhizome has been used as an aromatic stomachic in China. The chemical composition and biological activities of EO from *R. beesianus* rhizome were reported for the first time. Based on gas chromatography with flame ionization or mass selective detection (GC-FID/MS) results, the major constituents of EO were 1,8-cineole (47.6%), borneol (15.0%), methyleugenol (11.2%), and bornyl formate (7.6%). For bioactivities, EO showed a significant antibacterial activity against *Staphylococcus aureus*, *Enterococcus faecalis*, *Bacillus subtilis*, *Escherichia coli*, *Pseudomonas aeruginosa*, and *Proteus vulgaris* with the diameter of the inhibition zone (DIZ) (8.66–10.56 mm), minimal inhibitory concentration (MIC) (3.13–6.25 mg/mL), and minimal bactericidal concentration (MBC) (6.25–12.5 mg/mL). Moreover, EO (128 μg/mL) significantly inhibited the production of proinflammatory mediators nitric oxide (NO) (92.73 ± 1.50%) and cytokines tumor necrosis factor-α (TNF-α) (20.29 ± 0.17%) and interleukin-6 (IL-6) (61.08 ± 0.13%) in lipopolysaccharide (LPS)-induced RAW264.7 macrophages without any cytotoxic effect. Moreover, EO exhibited significant acetylcholinesterase (AChE) inhibitory activity (the concentration of the sample that affords a 50% inhibition in the assay (IC_50_) = 1.03 ± 0.18 mg/mL) and moderate α-glucosidase inhibition effect (IC_50_ = 11.60 ± 0.25 mg/mL). Thus, the EO could be regarded as a bioactive natural product and has a high exploitation potential in the cosmetics and pharmaceutical industries.

## 1. Introduction

Essential oils are volatile complex compounds with a strong odor that are formed by aromatic plants [1]. About 3000 essential oils have been produced by using at least 2000 plant species, of which nearly 300 kinds of essential oils have been used in health, perfume, cosmetic, agriculture, and food industries [2,3]. Additionally, essential oils have therapeutic uses in human medicine due to their antibacterial, anti-inflammatory, antinociceptive, antiviral, anticancer, vasodilatory, and penetration enhancing properties [4]. In recent years, due to the emergence of resistant strains of pathogens, the limitations of available antibiotics/drugs, and the side effects of synthetic drugs, people are encouraged to use essential oils as complementary and alternative therapies [5].

The Zingiberaceae family comprises approximately 52 genera and 1600 species and is mainly distributed in the tropical and subtropical regions [6,7]. Many species of the Zingiberaceae are rich in essential oil and cultivated for their various applications in dyes, spices, ornamental, cosmetics, medicine, and food industries [7,8]. The essential oils from the Zingiberaceae plants have been demonstrated to have multiple bioactivities, such as antimicrobial, anti-inflammatory, anticancer, antimutagenic, analgesic, anti-allergic, anti-ulcer, insecticidal, and immunomodulatory activities [5,8,9,10,11,12].

*Rhynchanthus* J. D. Hooker, a small genus in the family Zingiberaceae, has approximately four species and is mainly distributed in Myanmar, Indonesia, and Southern China [13,14]. *Rhynchanthus beesianus* W. W. Smith, a perennial herb with tuberous rhizomes is cultivated as an edible, medicinal, and ornamental plant in Myanmar and Southern China [14,15]. The rhizomes and tender leaves of *R. beesianus* are used as edible spices and vegetables in Yunnan Province, China. Its flower shape is peculiar, like a colorful and gorgeous brush, which can be used as fresh cut flowers or used for hanging cultivation. The rhizome has been used in Chinese folk medicine and as an aromatic stomachic, commonly known as *diangaoliangjiang*, to treat stomachache, backache, and indigestion [16,17,18]. Additionally, the EO from its rhizome has been used as an aromatic stomachic in China [17]. However, there is no study on the chemical composition and pharmacological activity of *R. beesianus*, which could impede its exploitation in the industry. Therefore, the purpose of this research is to study the chemical composition of the essential oil from *R. beesianus* rhizome and evaluate its antibacterial, anti-inflammatory and enzyme inhibitory activities.

## 2. Results and Discussion

### 2.1. Chemical Composition

The hydrodistillation of fresh rhizomes of *R. beesianus* yielded EO at 0.22% (*w*/*w*) on a fresh weight basis. A total of thirty-five chemical constituents were identified by gas chromatography with flame ionization detection (GC-FID) and GC-mass selective detection (MS), accounting for 98.3% of the EO composition (Table 1). As shown in Figure 1, the major constituents of *R. beesianus* EO were 1,8-cineole (47.6%), borneol (15.0%), methyleugenol (11.2%), bornyl formate (7.6%), camphene (3.4%), *α*-terpineol (2.7%), and *α*-pinene (2.5%).

### 2.2. Antibacterial Activity

The antibacterial activity of EO was qualitatively evaluated by the diameter of inhibition zone (DIZ) and quantitatively determined by the minimal inhibitory concentration (MIC) and minimal bactericidal concentration (MBC) values using streptomycin as the positive control (Table 2). The EO revealed broad-spectrum antibacterial activity with DIZ values ranging from 8.66 to 10.56 mm against Staphylococcus aureus (MIC = 6.25 mg/mL, MBC = 6.25 mg/mL), Enterococcus faecalis (MIC = 6.25 mg/mL, MBC = 6.25 mg/mL), Bacillus subtilis (MIC = 3.13 mg/mL, MBC = 6.25 mg/mL), Escherichia coli (MIC = 6.25 mg/mL, MBC = 12.50 mg/mL), Pseudomonas aeruginosa (MIC = 6.25 mg/mL, MBC = 12.50 mg/mL), Proteus vulgaris (MIC = 3.13 mg/mL, MBC = 6.25 mg/mL). The 1,8-cineole, as the most predominant component of R. beesianus EO, has been demonstrated to have significant antibacterial activity [19,20]. Recent studies have shown that borneol has an effective broad-spectrum antibacterial capability against Gram-positive, Gram-negative bacteria, and even multi-drug resistant bacteria via membrane disruption mechanism [21]. According to the study of Donadu et al., EO rich in 1,8-cineole and α-pinene showed significant antibacterial against Staphylococcus aureus and methicillin-resistant Staphylococcus aureus with minimum inhibitory concentrations (MIC) and minimum lethal concentration (MLC) values from 2 to 4% (*v*/*v*) [22]. Additionally, the antibacterial activity of other main components in R. beesianus EO, such as methyleugenol, camphene, α-terpineol, and α-pinene, have been demonstrated in the previous reports [23,24,25,26]. Therefore, the significant antibacterial property of R. beesianus EO can be attributed to these main components, and it can provide the natural antibacterial agents for the cosmetics and pharmaceutical industries.

### 2.3. Anti-inflammatory Activity

The anti-inflammatory activity of the EO from *R. beesianus* rhizome was investigated against lipopolysaccharide (LPS)-induced inflammation in RAW264.7 macrophages. The cytotoxicity of *R. beesianus* EO on L929 and RAW264.7 cells were evaluated using 3-[4,5-dimethylthiazol-2-yl]-2,5 diphenyl tetrazolium bromide (MTT) assay. Compared with the untreated control cells, *R. beesianus* EO showed no significant cytotoxicity on L929 and RAW264.7 cells at 16−128 μg/mL (Figure 2). Therefore, subsequent experiments were performed using EO concentrations of 16−128 μg/mL. Compared with the control group, RAW264.7 cells induced by LPS increased in size and became irregular in shape. RAW264.7 cells in the EO-treated group showed a relatively smooth surface compared to those in the LPS-induced group (Figure 3A). The accumulation of proinflammatory mediator nitric oxide (NO) in the culture supernatants were assayed by the Griess reaction using a colorimetric NO detection kit. Dexamethasone (DXM, 20 μg/mL) was used as a positive control. As shown in Figure 3B, compared with the control (1.46 ± 0.19 μM), stimulation with LPS alone resulted in a more than 13-fold increase in NO production (19.44 ± 1.88 μM). EO inhibited the production of NO in a dose-dependent manner. In particular, pretreatment with 128 μg/mL EO significantly decreased NO production by 92.73 ± 1.50% (2.84 ± 0.31 μM), which was equivalent to that of the positive control DXM (92.56 ± 0.70%, 2.80 ± 0.19 μM). Tumor necrosis factor-α (TNF-α) and interleukin-6 (IL-6) are two important pro-inflammatory cytokines, which play a key role in inflammatory disorders [27]. The levels of TNF-α and IL-6 in the culture supernatant were determined by enzyme-linked immunosorbent (ELISA) assay using an ELISA determination kit. As shown in Figure 3C, compared with the LPS group (3050.07 ± 4.04 pg/mL), EO significantly inhibited the secretion of TNF-*α* in LPS-induced RAW264.7 macrophages at doses of 64 μg/mL (2731.02 ± 1.80 pg/mL) and 128 μg/mL (2451.02 ± 3.52 pg/mL). The maximum inhibition rate of EO (20.29 ± 0.17% at 128 μg/mL) was comparable to that of DXM (21.34 ± 0.20% at 20 μg/mL). Additionally, compared with the LPS group (974.28 ± 4.15 pg/mL), EO effectively inhibited the secretion of IL-6 in RAW264.7 cells induced by LPS at doses of 16 μg/mL (803.81 ± 28.82 pg/mL), 32 μg/mL (738.81 ± 5.87 pg/mL), 64 μg/mL (550.48 ± 8.30 pg/mL), and 128 μg/mL (379.17 ± 0.31 pg/mL). In particular, the inhibitory ratios of EO at 64 μg/mL (43.50 ± 1.09%) and 128 μg/mL (61.08 ± 0.13%) were exceeded that of DXM (21.90 ± 0.14% at 20 μg/mL) (Figure 3D). 

The proinflammatory mediator (NO) and cytokines (TNF-*α* and IL-6) play key roles in the development of the inflammation process, and compounds that inhibit their overproduction may be good candidates in the treatment of many inflammation-related diseases [28]. According to the previous studies, the 1,8-cineole, as the most predominant constituent in *R. beesianus* EO, reduced LPS-induced inflammation by inhibiting TNF-*α* and IL-6 [29]. The anti-inflammatory properties of other main components in EO, such as borneol, methyleugenol, *α*-terpineol, and *α*-pinene, have been demonstrated in the previous reports [30,31,32,33]. Therefore, these main components could explain the significant anti-inflammatory effect of the *R. beesianus* EO. These results suggest that the *R. beesianus* EO can be used as a new source of natural anti-inflammatory molecules in the cosmetics and pharmaceutical industries.

### 2.4. Enzyme Inhibitory Activity

The inhibitory effect of *R. beesianus* EO on α-glucosidase, tyrosinase, acetylcholinesterase (AChE) and butyrylcholinesterase (BChE) were investigated and the results are shown in Table 3. 

α-Glucosidase inhibitors, such as acarbose, have been used to treat type 2 diabetes by reducing post-meal blood glucose and insulin levels [34]. Pre-diabetes is the preceding condition of diabetes and can lead to the development of diabetes in most cases. The treatment of pre-diabetes has shown great success in preventing the further progression of diabetes [35]. The lower doses of acarbose have been shown to have beneficial effects on pre-diabetes by delaying the absorption of carbohydrates in the intestine, but its use is limited by its side effects, such as flatulence, diarrhea, and abdominal distension [36,37]. Hence, it is very promising to screen natural products that prevent pre-diabetic from developing into type 2 diabetes by inhibiting the activity of α-glucosidase to reduce glucose absorption. EO exhibited moderate α-glucosidase inhibition effect (the concentration of the sample that affords a 50% inhibition in the assay (IC_50_ = 11.60 ± 0.25 mg/mL). The inhibitory activities of 1,8-cineole and methyleugenol against α-glucosidase were reported in the previous studies [38,39]. Hence, *R. beesianus* EO could be a natural source of potent α-glucosidase inhibitors.

Tyrosinase is a key enzyme which involved the catalysis of the only rate-limiting step in mammalian melanogenesis, so tyrosinase inhibitors are used to treat hyperpigmentation in the human skin [40,41]. As shown in Table 3, EO showed a weak inhibitory effect on tyrosinase (IC_50_ = 53.71 ± 4.89 mg/mL). 

The inhibition of cholinesterase can reduce the decomposition of acetylcholine to enhance cholinergic neurotransmission, which has become an effective treatment strategy for Alzheimer’s disease [42]. EO exerted the best AChE inhibitory effect with the lowest IC_50_ value of 1.03 ± 0.18 mg/mL. However, EO showed a weak inhibitory effect on BChE (IC_50_ = 104.22 ± 11.61 mg/mL). According to the previous studies, 1,8-cineole and *α*-pinene showed a strong AChE inhibitory activity and were found to be uncompetitive reversible AChE inhibitors [19,43,44]. Additionally, the significantly inhibited AChE activity of methyleugenol and *α*-terpineol has been demonstrated in the previous studies [44,45]. Therefore, the presence of these main components could explain the significant AChE inhibitory effect of EO. These results indicate that the *R. beesianus* EO can be used as a new source of AChE inhibitor in the pharmaceutical industry.

## 3. Materials and Methods

### 3.1. Plant Material

The fresh R. beesianus rhizome was collected from Guangxi Province of China in July 2019. The identity of this species was confirmed by Prof. Guoxiong Hu of Guizhou University. The voucher specimen (Voucher No: RB-20190711) was deposited at National & Local Joint Engineering Research Center for the Exploition of Homology Resources of Southwest Medicine and Food, Guizhou University.

### 3.2. Isolation of Essential Oil

The fresh, finely chopped rhizomes (2.0 kg) were subjected to hydrodistillation for 4 h using a Clevenger-type apparatus to obtain the essential oil (4.42 g, 0.22% *w*/*w*). EO was dried over anhydrous Na_2_SO_4_, filtered, and stored at 4 °C until further analysis.

### 3.3. Essential Oil Analysis and Identification

The EO was analyzed by an Agilent 6890 gas chromatograph (GC) equipped with a flame ionization detector (FID). Capillary column: HP-5MS (60 m × 0.25 mm, 0.25 μm film thickness). The injection volume was 1 μL and split injection was used (split ratio 1:20). The flow rate of carrier gas helium was set at 1 mL/min. The following GC oven temperature was used: kept at 70 °C (2 min), 2 °C per min to 180 °C (55 min), 10 °C per min to 310 °C (13 min), and held at 310 °C (4 min). The GC-MS analysis was carried out using an Agilent 6890 gas chromatograph equipped with an Agilent 5975C mass selective detector (Agilent Technologies Inc., CA, USA). GC column and parameters were the same as in GC-FID. The mass spectra operated in the mass range (*m*/*z* 29 to 500) and EI mode (70 eV). The interface temperature and ion source temperature were 280 °C and 230 °C, respectively. The relative percentage of chemical constituents was determined by the peak area. The retention index (RI) was determined by referring to a series of n-alkanes (C_8_–C_21_). The constituents of the EO were identified by comparison of their retention index and mass spectrum with those listed in NIST 2017 and Wiley 275 databases. 

### 3.4. Antibacterial Activity

The antibacterial capacity of EO was performed against six bacterial strains: *Staphylococcus aureus* ATCC 6538P, *Enterococcus faecalis* ATCC 19433, *Bacillus subtilis* ATCC 6633, *Escherichia coli* CICC 10389, *Pseudomonas aeruginosa* ATCC 9027, and *Proteus vulgaris* ACCC 11002.

The diameter of the inhibition zone (DIZ) was assayed according to our previously published disc agar diffusion method [46]. Briefly, The EO was dissolved with ethyl acetate (100 mg/mL). Streptomycin distilled water solution (100 μg/mL) was used as a positive control. The bacterial suspension (100 μL, 10^6^ CFU/mL) was evenly spread on the Mueller–Hinton agar medium. The filter paper discs (diameter 6 mm) containing EO or streptomycin solution (20 μL) were added and the diameter of inhibition zone (DIZ) was recorded after 24 h incubation at 37 °C.

The minimal inhibitory concentration (MIC) and minimal bactericidal concentration (MBC) values were evaluated by the microplate dilution method [46]. Briefly, the two-fold serially diluted sample solution (100 μL) and the bacterial suspensions (100 μL) were added to each well at the final density of 10^5^ CFU/mL. The 96-well plates were incubated for 24 h at 37 °C. Then, 20 μL of resazurin solution (0.1 mg/mL) was added to each well and incubated for 2 h in the dark at 37 °C. The MIC was determined as the minimum sample concentration without color change. For the determination of MBC value, the sample (10 μL) from the wells without color change was subcultured in a Mueller–Hinton agar plate for 24 h at 37 °C. MBC was defined as the minimum sample concentration without bacterial growth.

### 3.5. Anti-Inflammatory Activity

Murine macrophages RAW264.7 and murine fibroblast cells L929 were maintained at 37 °C with 5% CO_2_ atmosphere in Dulbecco’s modified eagle medium (DMEM) and Roswell Park Memorial Institute (RPMI) 1640 medium, respectively. Both media were supplemented with 2 mM glutamine, 10% fetal bovine serum, 100 U/mL penicillin, and 100 μg/mL streptomycin. The cytotoxic activity was determined by the MTT assay with slight modification [47]. The cell suspensions (100 μL) were added to each well at the density of 2 × 10^4^ cells per well and incubated for 24 h. Subsequently, 100 μL of the diluted EO solution was added and cultured for 24 h. Then, the MTT (3-[4,5-dimethylthiazol-2-yl]-2,5 diphenyl tetrazolium bromide) solution (10 μL, 5 mg/mL in PBS) was added. After 4 h of incubation, the supernate was discarded and 150 μL of DMSO was added to each well to dissolve formazan crystal. The absorbance was measured using a Varioskan Lux Multimode microplate reader (Thermo Fisher Scientific, Waltham, USA) at 490 nm to evaluate the cell viability and proliferation inhibitory ratio.

The RAW264.7 cell suspensions (100 μL) were added to each well at the density of 2 × 10^4^ cells per well and allowed to grow for 24 h. After removing the medium, the cells were incubated with 100 μL of two-fold serially diluted EO solution for 2 h. Then, 100μL lipopolysaccharide (LPS) was added to the final concentration of 1 μg/mL and incubated for another 24 h. Morphological changes of the treated RAW264.7 cells were recorded using an inverted microscope (Leica DMi8, Leica Microsystems, Germany). Subsequently, the supernatants were collected and centrifuged. The accumulation amount of NO in the supernatant was detected using a colorimetric NO detection kit following the manufacturer’s instructions (Nanjing Jiancheng Bioengineering Institute, Nanjing, China). The secretion of IL-6 and TNF-α was determined using respective ELISA determination kits according to the manufacturer’s instructions (Multi Sciences Biotech Co., Ltd., Hangzhou, China). Dexamethasone (DXM) (20 μg/mL) was used as a positive control.

### 3.6. Enzyme Inhibitory Activities

The α-glucosidase inhibitory activity was assayed according to the method previously described with slight modification [48]. EO solution (30 μL), phosphate buffer (60 μL, pH 6.8), and α-glucosidase solution (10 μL, 0.8 U/mL) were mixed in a 96-well plate and incubated at 37 °C for 15 min. Then, p-Nitrophenyl-α-D-glucopyranoside (p-NPG) solution (10 μL, 1 mM) was added and incubated at 37 °C. After 15 min, the reaction was stopped by adding Na_2_CO_3_ solution (80 μL, 0.2 M), and finally the absorbance was measured at 405 nm. Acarbose was used as a positive reference. The α-glucosidase inhibitory effect was expressed using IC_50_ values.

The inhibitory activity of tyrosinase was evaluated according to the method with L-tyrosine as a substrate [49]. EO solution (70 μL) and tyrosinase solution (100 μL, 100 U/mL) were mixed and added to each well. After 5 min incubation at 37 °C, 80 μL of L-tyrosine solution (5.5 mM) was added and incubated at 37 °C for 30 min. Then, the absorbance was recorded at 492 nm and arbutin was used as a positive control. The tyrosinase inhibitory activity was expressed using IC_50_ values.

The inhibitory effects of cholinesterase including acetylcholinesterase (AChE) and butyrylcholinesterase (BChE) were assayed according to Ellman’s method with marginal modification [50]. EO solution (50 μL) was mixed with AChE or BuChE solution (10 μL, 0.5 U/mL) in a 96-well plate and incubated at 4 °C for 10 min. Subsequently, acetylthiocholine iodide (ATCI) or butyrylthiocholine chloride (BTCl) solution (20 μL, 2 mM) and 5,5′-dithiobis-(2-nitrobenzoic acid) (DTNB) solution (20 μL, pH 8.0, 2 mM) were added and incubated at 37 °C for 30 min. After that, the absorbance was read at 405 nm and galanthamine was used as a positive control. The cholinesterase inhibitory effects were expressed using IC_50_ values.

### 3.7. Statistical Analysis

All experiments were performed independently at least three times, and the data were expressed as the mean ± standard deviation (SD). SPSS software (version 19.0) was used for statistical analysis. The significant difference between the two groups was compared by one-way analysis of variance (ANOVA) using Tukey’s multiple range tests (*P* < 0.05).

## 4. Conclusions

To our knowledge, the chemical composition, antibacterial, anti-inflammatory, and enzyme inhibitory activities of the essential oil from *R. beesianus* rhizome were reported for the first time. Thirty-five chemical constituents were identified and quantified with GC-FID/MS. In addition to the weak inhibitory effect of EO on tyrosinase and BChE, the *R. beesianus* EO showed significant antibacterial activity against *Staphylococcus aureus*, *Enterococcus faecalis*, *Bacillus subtilis*, *Escherichia coli*, *Pseudomonas aeruginosa*, *Proteus vulgaris*. Moreover, EO significantly inhibited the production of proinflammatory mediators (NO) and cytokines (TNF-α and IL-6) in LPS-induced RAW264.7 cells without any cytotoxic effect. Furthermore, EO revealed a moderate *α*-glucosidase inhibition effect and significant AChE inhibitory activity. Thus, the *R. beesianus* EO could be regarded as a source of bioactive products with high exploitation potential in the cosmetic and pharmaceutical industries.

## Figures and Tables

**Figure 1 molecules-26-00167-f001:**
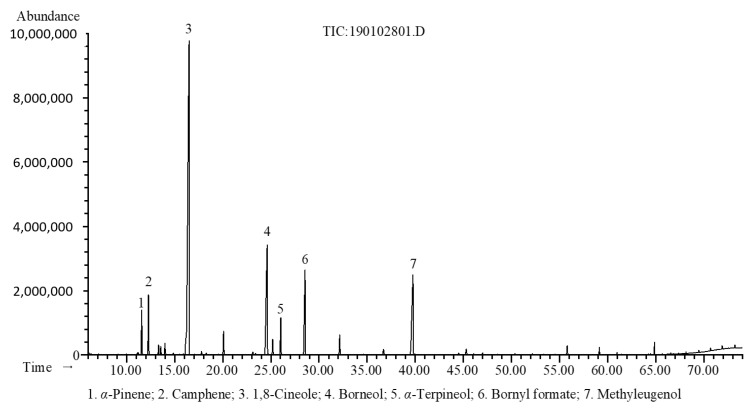
Gas chromatography with mass selective detection (GC-MS) chromatogram of *R. beesianus* rhizome essential oil (EO).

**Figure 2 molecules-26-00167-f002:**
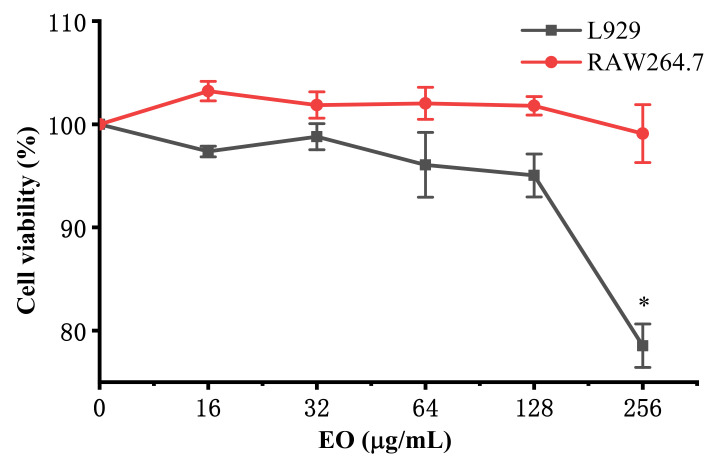
The effect of *R. beesianus* rhizome EO on the cell viability of RAW264.7 (murine macrophage cells) and L929 (murine fibroblast cells). L929 and RAW264.7 cells were pre-treated with different concentrations of *R. beesianus* EO for 24 h, and the cell viability was assayed by 3-[4,5-dimethylthiazol-2-yl]-2,5 diphenyl tetrazolium bromide (MTT) assay. The results were expressed as the percentage (%) of viable cells compared to the untreated cells and showed the mean ± standard deviation (SD). * *p* < 0.05, compared to the control group.

**Figure 3 molecules-26-00167-f003:**
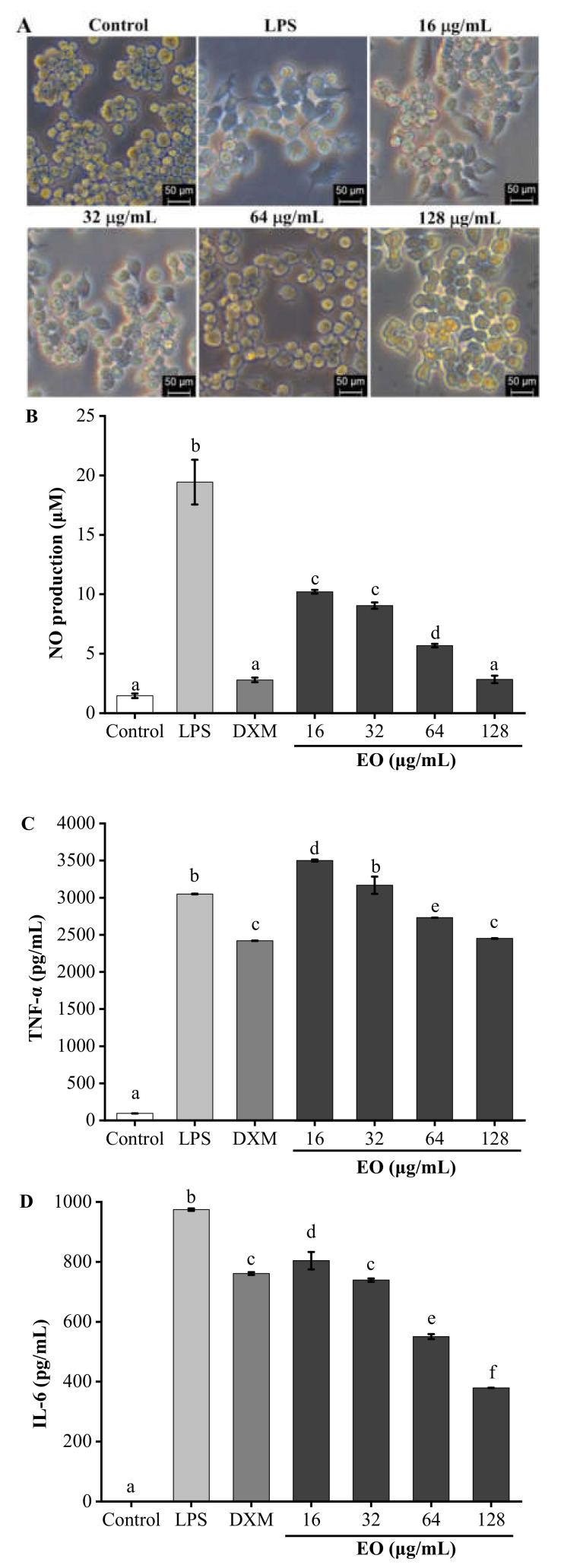
Effects of *R. beesianus* rhizome EO on the lipopolysaccharide (LPS)-induced morphology (200×) (**A**), nitric oxide (NO) production (**B**), and secretion of tumor necrosis factor-α (TNF-*α*) (**C**) and interleukin-6 (IL-6) (**D**) in RAW264.7 macrophages. (**A**) The RAW 264.7 macrophages were exposed to different EO concentrations and in the presence or absence of LPS (1 μg/mL). After 24 h of incubation, the images of RAW 264.7 cells were taken. (**B**) The RAW 264.7 cells were exposed to different EO concentrations and stimulated with LPS. Dexamethasone (DXM, 20 μg/mL) was used as a positive control. The accumulation of NO in the culture supernatants was assayed by the Griess reaction. (**C**, **D**) The RAW 264.7 cells were exposed to different EO concentrations and stimulated with LPS. The amount of TNF-*α* (**C**) and IL-6 (**D**) was determined using respective ELISA determination kits. The experiments were performed independently at least three times, and the data were expressed as the mean ± SD values. The different letters above bars indicate a significant difference (*p* < 0.05).

**Table 1 molecules-26-00167-t001:** Chemical composition of the essential oil from *R. beesianus* rhizome.

Compounds ^a^	RI ^b^	RI ^c^	RT (min) ^d^	% Area	Identification ^e^
Octane	800	800	7.076	0.1	MS, RI
Tricyclene	926	925	11.109	0.1	MS, RI
*α*-Thujene	928	929	11.205	0.1	MS, RI
*α*-Pinene	936	937	11.575	2.5	MS, RI
Camphene	952	952	12.275	3.4	MS, RI
Sabinene	976	974	13.32	0.6	MS, RI
*β*-Pinene	981	979	13.559	0.5	MS, RI
*β*-Myrcene	991	991	13.991	0.7	MS, RI
*α*-Phellandrene	1008	1005	14.844	0.1	MS, RI
*α*-Terpinene	1020	1017	15.483	tr ^e^	MS, RI
*p*-Cymene	1027	1022	15.926	0.1	MS, RI
1,8-Cineole	1038	1032	16.495	47.6	MS, RI
*β*-Ocimene	1047	1037	17.055	tr ^e^	MS, RI
*γ*-Terpinene	1061	1060	17.793	0.2	MS, RI
*trans*-4-Thujanol	1069	1070	18.279	0.1	MS, RI
Terpinolene	1092	1088	19.551	0.1	MS, RI
Linalool	1101	1099	20.085	1.5	MS, RI
(+)-2-Bornanone	1149	1144	23.128	0.2	MS, RI
Citronellal	1153	1152	23.4	0.1	MS, RI
Borneol	1173	1167	24.616	15.0	MS, RI
4-Terpineol	1182	1177	25.187	0.9	MS, RI
*α*-Terpineol	1195	1190	26.033	2.7	MS, RI
Bornyl formate	1234	1226	28.544	7.6	MS, RI
Bornyl acetate	1290	1285	32.155	1.3	MS, RI
Eugenol	1361	1357	36.701	0.4	MS, RI
Methyleugenol	1409	1402	39.765	11.2	MS, RI
*α*-Curcumene	1487	1483	44.524	0.1	MS, RI
Methylisoeugenol	1499	1495	45.296	0.5	MS, RI
*β*-Bisabolene	1513	1509	46.073	0.1	MS, RI
Sesquicineole	1519	1516	46.436	tr ^e^	MS, RI
*δ*-Cadinene	1529	1524	47.007	0.1	MS, RI
2-(3-Isopropenyl-4-methyl-4-vinylcyclohexyl)-2-propanol	1555	1549	48.574	0.1	MS, RI
(-)-Spathulenol	1586	1577	50.354	0.1	MS, RI
*β*-Eudesmol	1660	1649	54.505	0.1	MS, RI
Ambrial	1815	1809	60.969	0.1	MS, RI
Total identified				98.3	

^a^ Compounds were listed in order of their elution from an HP-5MS column. ^b^ Retention index (RI) on HP-5MS column, calculated using a homologous series of C_8_–C_21_ alkanes. ^c^ Retention index (RI) from NIST 2017 and Wiley 275 mass spectral databases. ^d^ Retention time (RT, min) on HP-5MS column. ^e^ Identification: MS, based on comparison with Wiley 275 and NIST 2017 MS databases; RI, based on comparison of calculated RI with those reported in Wiley 275 and NIST 2017 databases. ^e^ tr: trace (trace <0.01%).

**Table 2 molecules-26-00167-t002:** Antibacterial activity of *R. beesianus* rhizome EO.

Bacterial Strains ^a^	EO			Streptomycin		
	DIZ ^b^ (mm)	MIC ^c^ (mg/mL)	MBC ^c^ (mg/mL)	DIZ ^b^ (mm)	MIC ^c^ (μg/mL)	MBC ^c^ (μg/mL)
Gram positive						
*S. aureus*	9.78 ± 0.79	6.25	6.25	18.58 ± 0.41	0.78	1.56
*E. faecalis*	8.66 ± 0.80	6.25	6.25	7.34 ± 0.38	12.50	25.00
*B. subtilis*	10.54 ± 1.25	3.13	6.25	18.05 ± 1.56	0.39	0.78
Gram negative						
*E. coli*	9.01 ± 0.59	6.25	12.50	18.71 ± 0.64	0.78	1.56
*P. aeruginosa*	7.29 ± 0.14	6.25	12.50	9.97 ± 0.63	3.13	12.50
*P. vulgaris*	10.56 ± 0.92	3.13	6.25	15.71 ± 0.77	0.39	1.56

^a^ Bacterial strains: *Staphylococcus aureus* (ATCC 6538P), *Enterococcus faecalis* (ATCC 19433), *Bacillus subtilis* (ATCC 6633), *Escherichia coli* (CICC 10389), *Pseudomonas aeruginosa* (ATCC 9027), and *Proteus vulgaris* (ACCC 11002). ^b^ DIZ: The diameter of the inhibition zones (mm) including the diameter of the disk (6 mm). Sample solution: EO was diluted with ethyl acetate, at a concentration of 100 mg/mL (tested volume: 20 μL); Positive control: Streptomycin (tested volume: 20 μL, 100 μg/mL). ^c^ MIC: Minimal inhibitory concentration (μg/mL); MBC: Minimal bactericidal concentration (μg/mL).

**Table 3 molecules-26-00167-t003:** The enzyme inhibitory activity of *R. beesianus* rhizome EO.

Samples	Enzyme Inhibitory Activity (IC_50_, mg/mL) ^1^			
	α-Glucosidase	Tyrosinase	Acetylcholinesterase	Butyrylcholinesterase
EO	11.60 ± 0.25 ^a^	53.71 ± 4.89 ^a^	1.03 ± 0.18 ^a^	104.22 ± 11.61 ^a^
Acarbose	0.21 ± 0.01 ^b^			
Arbutin		0.24 ± 0.06 ^b^		
Galanthamine ^*^			0.46 ± 0.04 ^b^	5.68 ± 0.41 ^b^

^1^ IC_50_: The concentration of the sample that affords a 50% inhibition in the assay. ^a-b^ Different letters in the same column represent significant differences (*p* < 0.05). ^*^ Galanthamine: IC_50_ (μg/mL).

## Data Availability

The data presented in this study are available on request from the corresponding author.

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
