# Peer review of "Chemical Composition, Antibacterial, Anti-Inflammatory, and Enzyme Inhibitory Activities of Essential Oil from *Rhynchanthus beesianus* Rhizome"

_molecules, 2020, doi:10.3390/molecules26010167_

Round 1

Reviewer 1 Report

The topic is really interesting and trendy. The article may be accepted for the publishing with minor corrections.
See the attachment.

Author Response

Dear Reviewers:

  Thanks for your comments concerning our manuscript entitled “Chemical Composition, Antibacterial, Anti-inflammatory, and Enzyme Inhibitory Activities of Essential Oil from Rhynchanthus beesianus Rhizome” (Manuscript ID: molecules-1046570). Those comments are all valuable and very helpful for revising and improving our paper, as well as the important guiding significance to our researches. We have studied comments carefully and have made correction which we hope meet with approval. Revised portion are marked in red in the paper. The main corrections in the paper and the responds to the reviewer’s comments are as flowing:

Responds to the reviewer’s comments:

Suggestion: “The topic is really interesting and trendy. The article may be accepted for the publishing with minor corrections. See the attachment.

Response: We have modified according to the content of the attachment. Revised portions are marked in red in the paper. Thank you for these positive and constructive comments and suggestions.  

Reviewer 2 Report

Partition analysis of the paper:

1 - Introduction: must be reformed in the content and in the writing of the general part review the syntax of the topic

2- Discussion: deepen the aspect in the discussion from "1,8-cineole, borneol, methyleugenol,camphene, α-terpineol, and α-pinene, have been demonstrated in previous reports [7-11]." Delve into the aspect on these components in some newly identified essential oils like this one, and citing the following references:

PMID: 32570731 ; PMID: 29034738 ; PMID: 32338449

3 - Check the bibliographic entries in the text, some of which are non-compliant, review some entries in the bibliographic references and necessarily insert those referred to in point 2 for the purpose of my acceptance.

4 - Review the English grammar and in particular the applied scientific English: in particular, verbal tenses and syntax in the discussion.

Author Response

Dear Reviewers:

  Thanks for your comments concerning our manuscript entitled “Chemical Composition, Antibacterial, Anti-inflammatory, and Enzyme Inhibitory Activities of Essential Oil from Rhynchanthus beesianus Rhizome” (Manuscript ID: molecules-1046570). Those comments are all valuable and very helpful for revising and improving our paper, as well as the important guiding significance to our researches. We have studied comments carefully and have made correction which we hope meet with approval. Revised portion are marked in red in the paper. The main corrections in the paper and the responds to the reviewer’s comments are as flowing:

Responds to the reviewer’s comments:

Suggestion 1:“Introduction: must be reformed in the content and in the writing of the general part review the syntax of the topic.

Response 1: We have reformed the content and the writing of the “Introduction”. Revised portions are marked in red in the paper. The main corrections in the revised manuscript are as follows:

Current added contents:

In page 1 and page 2:

Essential oils are volatile complex compounds with a strong odor that are formed by aromatic plants [1]. About 3000 essential oils have been produced by using at least 2000 plant species, of which nearly 300 kinds of essential oils have been used in health, perfume, cosmetic, agriculture, and food industries [2,3]. Additionally, essential oils have therapeutic uses in human medicine due to their antibacterial, anti-inflammatory, antinociceptive, antiviral, anticancer, vasodilatory, and penetration enhancing properties [4]. In recent years, due to the emergence of resistant strains of pathogens, the limitations of available antibiotics/drugs, and the side effects of synthetic drugs, people are encouraged to use essential oils as complementary and alternative therapies [5].

The Zingiberaceae family comprises approximately 52 genera and 1600 species and mainly distributes in the tropical and subtropical regions [6,7]. Many species of the Zingiberaceae are rich in essential oil and cultivated for their various applications in dyes, spices, ornamental, cosmetics, medicine, and food industries [7,8]. The essential oils from the Zingiberaceae plants have been demonstrated to have multiple bioactivities, such as antimicrobial, anti-inflammatory, anticancer, antimutagenic, analgesic, anti-allergic, anti-ulcer, insecticidal, and immunomodulatory activities [5, 8-12].

In page 9 and page 10 (References):

  1. Bakkali, F.; Averbeck, S.; Averbeck, D.; Idaomar, M. Biological effects of essential oils–a review. Food Chem. Toxicol. 2008, 46, 446–475.
  2. Burt,Essential oils:their antibacterial properties and potential applications in foods—a review.Int. J. Food Microbiol. 2004, 94, 223–253.
  3. Trong Le, N.; Viet Ho, D.;QuocDoan, T.; Tuan Le, A.; Raal, A.; Usai, D.; Madeddu, S.; Marchetti, M.; Usai, M.; Rappelli, P.;Diaz,;Zanetti,S.;ThiNguyen,H.;Cappuccinelli,P.;Donadu,M.G.InVitro Antimicrobial Activity of Essential Oil Extracted from Leaves of Leoheo domatiophorusChaowasku, D.T. Ngo and H.T. Le in Vietnam. Plants 20209, 453.
  4. Buchbauer, G.; Bohusch, R. Biological activities of essential oils: an update.InHandbook of Essential Oils: Science, Technology, and Applications; Baser, K.H.C., Buchbauer, , Eds.; CRC Press/Taylor & Francis Group: Boca Raton, USA, 2010; pp. 281–321.
  5. Raut,S; Karuppayil,S.M.A status review on the medicinal properties of essential oils. Ind. Crop. Prod. 2014, 62, 250–264.
  6. The Plant List. Version 1. Available online: http://www.theplantlist.org/1.1/browse/A/Zingiberaceae/ (accessed on 25 December 2020).
  7. Jantan, I.B.; Yassin, M.S.M.; Chin, C.B.; Chen, L.L.; Sim, N.L. Antifungal activity of the essential oils of nine Zingiberaceae species.  Biol. 2003, 41, 392–397.
  8. Tan, J.W.; Israf, D.A.; Tham, C.L. Major bioactive compounds in essential oils extracted from the rhizomes of Zingiber zerumbet (L) Smith: A mini-review on the anti-allergic and immunomodulatory properties. Pharmacol. 2018, 9, 652.
  9. Mahboubi, M. Zingiber officinale essential oil, a review on its composition and bioactivity. Clin. Phytosci., 2019, 5, 6.
  10. Balaji, S.; Chempakam, B. Anti-bacterial Effect of Essential Oils Extracted from Selected Spices of Zingiberaceae. Prod. J. 2018, 8, 70–76.
  11. Phukerd, U.; Soonwera, M. Larvicidal and pupicidal activities of essential oils from Zingiberaceae plants against Aedes aegypti (Linn.) and Culex quinquefasciatus Say mosquitoes. Southeast Asian J. Trop Med. Public Health. 2013, 44, 761–
  12. Tewtrakul, S.; Subhadhirasakul, S. Anti-allergic activity of some selected plants in the Zingiberaceae family.  Ethnopharmacol. 2007, 109, 535–538.

Suggestion 2:“Discussion: deepen the aspect in the discussion from "1,8-cineole, borneol, methyleugenol, camphene, α-terpineol, and α-pinene, have been demonstrated in previous reports [7-11]." Delve into the aspect on these components in some newly identified essential oils like this one, and citing the following references:PMID: 32570731 ; PMID: 29034738 ; PMID: 32338449

Response 2: We have deepened the discussion on the antibacterial activity of R. beesianus EO and cited the following references: PMID: 32570731; PMID: 29034738; PMID: 32338449. Revised portions are marked in red in the paper. The main corrections in the revised manuscript are as follows:

Current added contents:

In page 4: The 1,8-cineole, as the most predominant component of R. beesianus EO, has been demonstrated to have significant antibacterial activity [19,20]. Recent studies have shown that borneol has an effective broad-spectrum antibacterial capability against Gram-positive, Gram-negative bacteria, and even multi-drug resistant bacteria via membrane disruption mechanism [21]. According to the study of Donadu et al., EO rich in 1,8-cineole and α-pinene showed significant antibacterial against Staphylococcus aureus and methicillin-resistant Staphylococcus aureus with minimum inhibitory concentrations (MIC) and minimum lethal concentration (MLC) values from 2 to 4% (v/v) [22].

In page 10 (References):

  1. Kahkeshani, N.; Hadjiakhoondi, A.; Navidpour, L.; Akbarzadeh, T.; Safavi, M.; Karimpour-Razkenari, E.; Khanavi, M. Chemodiversity of Nepeta menthoides Boiss. & Bohse. essential oil from Iran and antimicrobial, acetylcholinesterase inhibitory and cytotoxic properties of 1,8-cineole chemotype. Nat. Prod. Res. 2018, 32, 2745–2748.
  2. Yang, L.; Zhan, C.; Huang, X.; Hong, L.; Fang, L.; Wang, W.; Su, J. Durable Antibacterial Cotton Fabrics Based on Natural Borneol-Derived Anti-MRSA Agents. Adv. Healthcare Mater. 2020, 9, 2000186.
  3. Donadu, M.G.; Trong Le, N.; Viet Ho, D.; Quoc Doan, T.; Tuan Le, A.; Raal, A.; Usai, M.; Marchetti, M.; Sanna, G.; Madeddu, S.; Rappelli, P.; Diaz, N.; Molicotti, P.; Carta, A.; Piras, S.; Usai, D.; Thi Nguyen, H.; Cappuccinelli, P.; Zanetti, S. Phytochemical Compositions and Biological Activities of Essential Oils from the Leaves, Rhizomes and Whole Plant of Hornstedtia bellaŠkorničk. Antibiotics 2020, 9, 334.

Suggestion 3:“Check the bibliographic entries in the text, some of which are non-compliant, review some entries in the bibliographic references and necessarily insert those referred to in point 2 for the purpose of my acceptance.

Response 3: We have checked and made revision the bibliographic and entries bibliographic references. Besides, we have cited the following references: PMID: 32570731; PMID: 29034738; PMID: 32338449. Revised portions are marked in red in the paper. Thank you for these positive and constructive comments and suggestions. 

Suggestion 4:“Review the English grammar and in particular the applied scientific English: in particular, verbal tenses and syntax in the discussion.

Response 4: We have tried our best to revise the English grammar and improve the discussion part, and have made revision which marked in red in the paper. Thank you for these positive and constructive comments and suggestions.

Reviewer 3 Report

Comments to author

          The manuscript by Zhao et al, has described for the first time the chemical composition, antibacterial, anti-inflammatory, and enzyme inhibitory activities of the essential oil from  Rhynchanthus beesianus rhizome. In my opinion, the article is well written and presented in a continuous manner, where results are clearly articulated. This is a demonstration of the complete quality of the work and presentation. This study provides fundamental insight into the composition of essential oil from Rhynchanthus beesianus and its use and also represents a template for future investigations. As such, I think it will be of substantial and broad interest to the molecule's readership and therefore should be published after addressing some minor points related to editing or proofreading provided below.

Minor comments:

  1. Authors should work more on explaining the legends for all figures putting more information.
  2. The author should expand all the abbreviations used in the paper at its first appearance to help the readers.
  3. For figure 3, panel A, include the scale bars for images and mention how the images were taken in the figure legend.
  4. For Figure 3, panel B, Y=axis represents NO ( nitric oxide) or Nitrite ) (No2)?

Author Response

Dear Reviewers:

  Thanks for your comments concerning our manuscript entitled “Chemical Composition, Antibacterial, Anti-inflammatory, and Enzyme Inhibitory Activities of Essential Oil from Rhynchanthus beesianus Rhizome” (Manuscript ID: molecules-1046570). Those comments are all valuable and very helpful for revising and improving our paper, as well as the important guiding significance to our researches. We have studied comments carefully and have made correction which we hope meet with approval. Revised portion are marked in red in the paper. The main corrections in the paper and the responds to the reviewer’s comments are as flowing:

Responds to the reviewer’s comments:

Suggestion 1:“Authors should work more on explaining the legends for all figures putting more information.

Response 1: We have work more on explaining the legends for all figures and provided more information. Revised portions are marked in red in the paper. Thank you very much for your comments and suggestions.

The main modification contents are as follows:

In page 5, lines 117-136:

The accumulation of proinflammatory mediator nitric oxide (NO) in the culture supernatants were assayed by the Griess reaction using a colorimetric NO detection kit. Dexamethasone (DXM, 20 μg/mL) was used as a positive control. As shown in Figure 3B, compared with the control (1.46 ± 0.19 μM), stimulation with LPS alone resulted in a more than 13-fold increase in NO production (19.44 ± 1.88 μM). EO inhibited the production of NO in a dose-dependent manner. In particular, pretreatment with 128 μg/mL EO significantly decreased NO production by 92.73 ± 1.50% (2.84 ± 0.31 μM), which was equivalent to that of the positive control DXM (92.56 ± 0.70%, 2.80 ± 0.19 μM). TNF-α and IL-6 are two important pro-inflammatory cytokines, which play a key role in inflammatory disorders [27]. The levels of TNF-α and IL-6 in the culture supernatant were determined by enzyme-linked immunosorbent (ELISA) assay using an ELISA determination kit. As shown in Figure 3C, compared with the LPS group (3050.07 ± 4.04 pg/mL), EO significantly inhibited the secretion of TNF-α in LPS-induced RAW264.7 macrophages at doses of 64 μg/mL (2731.02 ± 1.80 pg/mL) and 128 μg/mL (2451.02 ± 3.52 pg/mL). The maximum inhibition rate of EO (20.29 ± 0.17% at 128 μg/mL) was comparable to that of DXM (21.34 ± 0.20% at 20 μg/mL). Additionally, compared with the LPS group (974.28 ± 4.15 pg/mL), EO effectively inhibited the secretion of IL-6 in RAW264.7 cells induced by LPS at doses of 16 μg/mL (803.81 ± 28.82 pg/mL), 32 μg/mL (738.81 ± 5.87 pg/mL), 64 μg/mL (550.48 ± 8.30 pg/mL), and 128 μg/mL (379.17 ± 0.31 pg/mL). In particular, the inhibitory ratios of EO at 64 μg/mL (43.50 ± 1.09%) and 128 μg/mL (61.08 ± 0.13%) were exceeded that of DXM (21.90 ± 0.14% at 20 μg/mL) (Figure 3D).

In page 6, lines 156-166:

Figure 3. Effects of R. beesianus rhizome EO on the LPS-induced morphology (200×) (A), NO production (B), and secretion of TNF-α (C) and IL-6 (D) in RAW264.7 macrophages. (A) The RAW 264.7 macrophages were exposed to different EO concentrations and in the presence or absence of LPS (1 μg/ml). After 24 h of incubation, the images of RAW 264.7 cells were taken. (B) The RAW 264.7 cells were exposed to different EO concentrations and stimulated with LPS. Dexamethasone (DXM, 20 μg/mL) was used as a positive control. The accumulation of NO in the culture supernatants was assayed by the Griess reaction. (C, D) The RAW 264.7 cells were exposed to different EO concentrations and stimulated with LPS. The amount of TNF-α (C) and IL-6 (D) was determined using respective ELISA determination kits. The experiments were performed independently at least three times, and the data were expressed as the mean ± SD values. The different letters above bars indicate a significant difference (p<0.05).

Suggestion 2:“The author should expand all the abbreviations used in the paper at its first appearance to help the readers.

Response 2: We are sorry for such obvious mistakes. We have expanded all the abbreviations used in the paper at their first appearance and have made revision which marked in red in the paper. Thank you very much for your comments and suggestions.

Suggestion 3:“For figure 3, panel A, include the scale bars for images and mention how the images were taken in the figure legend

Response 3: We are sorry for such obvious mistakes. We have added the scale bars for figure 3A, and mentioned how the images were taken in the figure legend expanded (in page 6, lines 157-159). Revised portions are marked in red in the paper. Thank you very much for your comments and suggestions.

Suggestion 4:“For Figure 3, panel B, Y=axis represents NO (nitric oxide) or Nitrite (No2)?

Response 4: We are sorry for such obvious mistakes. For figure 3, panel B, Y=axis represents NO production. We have revised the figure 3B in the paper. Thank you very much for your comments and suggestions.

Round 2

Reviewer 2 Report

Corrections were made successfully.